



# Quantifying the impact of synoptic weather types and patterns on energy fluxes of a marginal snowpack

Andrew Schwartz[1], Hamish McGowan[1], Alison Theobald[2], Nik Callow[3]

[1]Atmospheric Observations Research Group, University of Queensland, Brisbane, 4072, Australia
[2]Department of Environment and Science, Queensland Government, Brisbane, 4072, Australia
[3]School of Agriculture and Environment, University of Western Australia, Perth, 6009, Australia

*Correspondence to:* Andrew J. Schwartz (Andrew.Schwartz@uq.edu.au)

**Abstract.**

Synoptic weather patterns are investigated for their impact on energy fluxes driving melt of a marginal snowpack in the Snowy Mountains, southeast Australia. K-means clustering applied to ECMWF ERA-Interim data identified common synoptic types and patterns that were then associated with in-situ snowpack energy flux measurements. The analysis showed that the largest contribution of energy to the snowpack occurred immediately prior to the passage of cold fronts through increased sensible heat flux as a result of warm air advection (WAA) ahead of the front. Shortwave radiation was found to be the dominant control on positive energy fluxes when individual synoptic weather types were examined. As a result, cloud cover related to each synoptic type was shown to be highly influential on the energy fluxes to the snowpack through its reduction of shortwave radiation and reflection/emission of longwave fluxes. This research is an important step towards understanding changes in surface energy flux as a result of shifts to the global atmospheric circulation as anthropogenic climate change continues to impact marginal winter snowpacks.

## 1 Introduction

### 1.1 Synoptic weather influences on snowpack processes

Water generated in mountainous regions is a commodity that over 50% of the world's population depends on for daily life (Beniston, 2003). Arguably, the most important role in the generation and regulation of these water resources is that of montane snowpacks. These have been referred to as "water towers" (Viviroli et al., 2007) due to their capabilities for storage and slow releases of meltwater. Many snowpacks are undergoing reductions in spatial and temporal extent as a result of anthropogenic climate change (Pachauri et al., 2014). Understanding the physical drivers of snowpack ablation, including synoptic-scale influences, is critical to help assess future water resource availability in mountainous regions as climate change continues.

Snowfall has been related to synoptic weather types in numerous studies globally including in Athens (Prezerakos and Angouridakis, 1984), the central and eastern United States (Goree and Younkin, 1966), the Tibetan Plateau (Ueno, 2005), Budapest (Bednorz, 2008a), and the central European lowlands (Bednorz, 2011). However, work on relationships between snowmelt and synoptic weather types is relatively scarce. Bednorz (2008b) identified increased air temperature and rain-on-snow events as causes for rapid snowmelt (> 5 cm day$^{-1}$) in the Polish-German lowlands as a result of west-southwest airflows over Central Europe during positive phases of the North Atlantic Oscillation (NAO). Similar work has been conducted in North America by Grundstein and Leathers (1998) who were able to identify three main synoptic weather types responsible for significant snowmelt events on the northern Great Plains, all of which included cyclonic influence with different low pressure centre locations and warm air advection to the region. While some knowledge exists on synoptic drivers of snowpack ablation,



further research is needed to understand synoptic effects on ablation processes over snowpacks with varying
characteristics.
Marginal snowpacks are characterised by high snow density and internal temperatures, making them susceptible
to melt from energy input throughout much of the season and particularly sensitive to even subtle shifts in
available energy. Anthropogenic climate change has led to changes in snowpack and precipitation properties
globally (Adam et al., 2009;Stewart, 2009) and regions that have been historically categorized as having lower
temperatures have begun developing marginal characteristics as temperatures increase. However, research related
to synoptic influences on the surface energy balance over marginal snowpacks as defined by Bormann et al. (2013)
are rare. Hay and Fitzharris (1988) studied the influence of different synoptic weather types on glacier ablation
and snowpack melt, while Neale and Fitzharris (1997) used surface energy flux measurements to determine which
synoptic types resulted in highest ablation in the Southern Alps, New Zealand. These studies found net radiation
was the dominant term in ablation, but also noted that the contributions made by each term varied largely
depending on the synoptic type and its meteorology. A common characteristic between these studies and others
in various regions is that they focused primarily on the surface meteorology for synoptic classifications rather
than multiple level analysis, which enables insight to the potential influence of mid and upper-level atmospheric
conditions on surface – atmosphere energy exchanges. Regardless, no analysis at any level exists on synoptic type
influence on snowpack ablation within Australia.
**1.2    The Australian snowpack**
Characteristics of the snowpack in the Australian Alps have been examined in a number of studies with focus on
spatial and temporal snow cover variability (Budin, 1985;Duus, 1992), influence on catchment hydrology (Costin
and Gay, 1961), the energetics of snowpack melt (Bilish et al., 2018), and isotopic composition of precipitation
(Callow et al., 2014). Given observed declines in snow cover, climate change has become a central focus of this
research (Chubb et al., 2011;Hennessy et al., 2008;Nicholls, 2005;Reinfelds et al., 2014;Whetton et al., 1996) as
any changes to energy flux over the region will significantly impact the already marginal snowpack. Hennessy et
al. (2008) showed that future projections for the Australian snowpack predict reductions in annual areal snow
cover of 10-39% by 2020 and 22-85% by 2050. Observations indicate that reduction in snow cover is already
occurring with shortened annual periods of wintertime precipitation. Nicholls (2005) found reductions of 10%
and 40% in the maximum snow depth and snow depth at the first October measurement respectively from 1962
to 2002. In addition, wintertime precipitation was shown to have reduced by an average of 43% in high elevation
regions from 1990 to 2009 (Chubb et al., 2011), though much of this could have been due to several severe
droughts that occurred during the study period. Fiddes et al. (2015) showed that snowfall, snow accumulation,
and snow depth were highly correlated with temperature and that warming, as a result of climate change, could
lead to further reductions in the southeast Australia (SEA) snowpack. The importance of the water generated in
the Australian Alps, reduction in wintertime precipitation amounts and frequency, and high spatiotemporal
variability of snow accumulation and ablation (Budin, 1985) warrants an understanding of the energetics of
Australia's snowpack as they pertain to the influences of shifting synoptic-scale circulations.
**1.3    Synoptic weather types and trends in the Australian Alps**
The Australian Alps is a marginal snowpack environment (Bilish et al., 2019;Bilish et al., 2018), where
precipitation is crucial to agriculture, the generation of hydroelectric energy, and recreation and was estimated to





be worth $9.6 billion per year in 2005 (Worboys and Good, 2011). A maximum in precipitation in the Australian
Alps typically occurs during the cooler months of June to September when it falls as snow at elevations above
1400 m, and accounts for twice as much precipitation as during the warmer periods of the year (Chubb et al.,
2011). While the snowpack typically exists for relatively short periods compared to those of other regions where
winter temperatures are lower and higher snowfall amounts occur such as parts of the European Alps and Rocky
Mountains, USA, it is still a vital resource for SEA.
Synoptic weather types in Australia have been changing in recent decades in response to the impact of climate
change on background climate states (Theobald et al., 2016;Hope et al., 2006). For example, increases in daily
maximum temperature and reductions in precipitation during autumn and winter have been noted in SEA as a
result of anomalously high surface pressure during positive periods of the Southern Annular Mode (SAM)
(Hendon et al., 2007). Cai et al. (2005) also showed an increase in SAM value as a response to all global warming
experiments using the CSIRO Mark 3 climate model indicating a further poleward shift in the location of synoptic
systems. However, it has been suggested that the SAM accounts for a relatively small portion of seasonal rainfall
variability in Australia and other larger impacts on synoptic weather from other sources are likely (Meneghini et
al., 2007).
Significant work has been conducted on identification of patterns and trends in Australian synoptic climatology
as it pertains to precipitation variability (Theobald et al., 2016;Chubb et al., 2011;Pook et al., 2014;2010;Pook et
al., 2006;2012). However, impacts on surface energy fluxes as a result of synoptic types have not been explored
as they have in other regions. The objective of this study is to identify the synoptic weather types that contribute
the highest amounts of energy to the Australian snowpack. This is accomplished through: 1) the identification and
classification of common synoptic types during periods of homogeneous snow cover, 2) attribution of snowpack
energy flux characteristics to each synoptic type, and 3) construction of energy balance patterns as they pertain to
common synoptic patterns/progressions.
**2      Methods**
**2.1      Study site and climate**
Energy flux measurements were made 16 km west of Lake Jindabyne at the Pipers Creek catchment headwaters
(36.417°S, 148.422°E) at an elevation of 1828 m in the Snowy Mountains, Kosciuszko National Park, New South
Wales (NSW), Australia (Figure 1). The catchment is classified as sub-alpine and contains grasslands, sub-alpine
bogs, and sub-alpine woodland (Gellie, 2005). The surrounding areas contain a mixture of living and dead
*Eucalyptus pauciflora* (Snow Gum) trees and open grassland areas with fens and alpine bogs. Many of the Snow
Gums were impacted by fire in 2003, and have experienced slow regrowth. The area's mixed characteristics of
forested and open grasslands with alpine wetlands within the Pipers Creek study catchment and immediately
surrounding the flux tower site used in this study are representative of those found throughout the Australian Alps.
The Snowy Mountains are characterized by relatively mild weather conditions compared to other mountain
ranges. Winter temperatures are typically around 0°C with mean low temperatures during July (the coldest month)
at -5°C and mean high temperatures between 2 to 4°C (Bureau of Meteorology, 2018b) that readily allow for melt
of the snowpack. As such, snowpack properties in the catchment are consistent with those of maritime snowpacks





that are associated with basal melting, high temperatures, and high wind speeds (Sturm et al., 1995;Bilish et al.,
118  2018).

The site chosen at the Pipers Creek catchment headwaters contains alpine bog and Eucalypt woodland that are
"the two most common types in the broader region, together representing 47% of the total area above 1400-m
elevation" (Bilish et al., 2018, p. 3839). Gellie (2005) showed that the *E. pauciflora* woodland was present in five
of the fifteen dominant vegetation formations that covers 57% of area within the broader region, while Alpine
grassland/bog (including herb fields) accounts for another 8%. Additionally, the study site was located at 1828 m
in the middle of the 1400-2228 m elevation range in the Australian Alps that typically has snowfall during the
winter allowing for measurements that apply broadly to other elevations. As with all single-site studies, there will
be some uncertainty when applying the energy balance to the wider area of the Australian Alps. However, this
should be reduced as the measurements made include surface types common in the wider region and were towards
the middle of the elevations that those conditions occur.
**2.2  Instrumentation**
The Pipers Creek site (Figure 2) was established on 10 June 2016 and collected data for the 2016 and 2017 winter
seasons. The site consisted of a Campbell Scientific eddy covariance (EC) system to measure fluxes of latent ($Q_e$)
and sensible ($Q_h$) heat at 10 Hz at a height of 3.0 m above ground level (AGL). A Kipp and Zonen CNR4
radiometer (3.0 m AGL) was used to measure incoming and outgoing shortwave (K) and longwave (L) radiation
to allow for comparisons of all radiation components rather than simply net all-wave radiation (Q*). Ambient air
temperature and relative humidity were measured at the top of the mast by a Vaisala HMP155 probe at ~3.1 m
above ground level. A Hukseflux heat flux plate measured ground heat flux ($Q_g$) at a depth of 5 cm and was placed
approximately 0.5 m from the centre of the mast to minimize any influence the mast could have on snow
accumulation above the sensor. Surface temperatures were monitored using an Apogee Instruments SI-111
infrared radiometer at approximately 2 m from the centre of the mast. Details on the instruments used for each
measurement are shown in Table 1.
Precipitation data from an ETI Instrument Systems NOAH II weighing gauge located approximately 1 km to the
northwest of the energy balance site at elevation of 1761 m was supplied by Snowy Hydro Limited (SHL). A 6 m
diameter DFIR shield was used around the gauge in order to prevent wind-related under-catch of snowfall
(Rasmussen et al., 2012), and was additionally sheltered by vegetation to the west.
**2.3  Identification of snow cover periods**
Homogeneous snow cover is crucial to accurate measurement and analysis of snowpack energy balance (Reba et
al., 2009). Snow cover was considered to be homogeneous when no grass or bush was protruding from the snow
surface with the exception of distant patches of *E. pauciflora* trees. Periods with homogeneous snow cover were
determined using data from the Pipers Creek instrumentation site and were cross referenced to manual snow
measurements made at the Spencers Creek Snow Course 6.6 km northwest of the Pipers Creek field site (Snowy
Hydro Ltd, 2018). Periods with surface temperatures above 1.5°C as measured by the SI-111 infrared radiometer
that did not correspond to rain-on-snow events and periods with albedo measurements less than 0.40 (Robock,
1980) were considered to have heterogeneous snow cover and were eliminated.



### 2.4 Synoptic classification of snow cover days

Synoptic weather type classification of homogeneous snow cover days was conducted using synoptic typing methods adapted from Theobald et al. (2015). European Centre for Medium-Range Weather Forecasts (ECMWF) ERA-Interim reanalysis data (Dee et al., 2011) with a 0.75° X 0.75° resolution was obtained for each day from 10 June 2016 through 31 October 2017. This date range was chosen to ensure inclusion of all potential dates with snow cover during the 2016 and 2017 snow seasons after the initial instrument tower installation on 10 June 2016. Variables included in the reanalysis data consisted of mean daily values of Mean Sea Level Pressure (MSLP); temperature and relative humidity at 850, 700, 500, and 250 hPa; wind vectors at 10 m AGL, 850, 700, 500, and 250 hPa; and 1000-500 hPa geopotential heights. The domain of the included variables was limited to 20°S - 46°S and 120°E -160°E, ensuring coverage of synoptic scale systems affecting the Australian Alps.

Focus was placed on analysis of temperature ($T_d$) and relative humidity ($RH$) values because of their impact on $Q_e$, $Q_h$, and radiative fluxes (Reba et al., 2009;Ruckstuhl et al., 2007;Allan et al., 1999;Webb et al., 1993). Relative humidity values at 850, 700, and 500 hPa were used to investigate the potential influence of cloud cover. MSLP and wind vector analysis at the 850, 700, 500, and 250 hPa levels allowed for the identification of $T_d$ and $RH$ advection (Pook et al., 2006) into the Australian Alps. Thickness between 1000-500 hPa was used to determine frontal positions relative to the Australian Alps (Pook et al., 2006) and accordingly the Pipers Creek field site.

The method used for synoptic comparison of energy flux characteristics was adopted from the approach of similar types of studies (Theobald et al., 2016;Theobald et al., 2015;Chubb et al., 2011;Neale and Fitzharris, 1997) and used "days" as the temporal period for analysis. "Days", periods lasting twenty-four hours from 00Z to 23:59Z, were considered optimal to determine radiative flux characteristics (diurnal radiation cycle) that may be missed on smaller time scales. Similarly, the use of days allows for determination of short-term energy fluxes that can also be easily compared over several months, thus being most appropriate for the entire snow season.

Days within the ERA-Interim data that matched snow cover days were extracted and analysed using the k-means clustering algorithm developed by Theobald et al. (2015). The algorithm was tested for 1-20 clusters and an elbow plot of the cluster distances was used to identify the optimum number of clusters (Theobald et al., 2015), which was seven. The identification of an elbow in the plot at seven clusters indicates a reduction to the benefit of adding additional clusters as the sum of distances for additional clusters fails to yield significant reductions beyond that point (Wilks, 2011).

Clustering of the synoptic conditions for each day was verified through manual analysis of MSLP and 500 hPa charts from the Australian Bureau of Meteorology (BOM) (Bureau of Meteorology, 2018). Cloud cover for each type was investigated and verified through the use of visible band Himawari-8 satellite data (https://www.ncdc.noaa.gov/gibbs/) at 03:00 UTC (13:00 local time) with one of three categories assigned to each day studied; 1) no cloud cover, 2) partial cloud cover, or 3) complete cloud cover. Cloud cover was investigated at midday to avoid misclassification due to short-lived clouds that appear over the area during the dawn and dusk periods.

Manual verification of the k-means clustering algorithm using BOM synoptic charts identified four days (2.45%) out of the 163 classified during the 2016 and 2017 seasons that had been classified incorrectly and they were subsequently moved to their correct synoptic type. Three of the four misclassified days were early (7 June 2016)



or late (19 and 22 September 2016) in the snowpack seasons with the fourth occurring in the middle of winter on
31 July 2017. Synoptic characteristics from these days tended to be complicated with no discernible dominant
features that matched those of classified types. This is likely due to shifting synoptic conditions between seasons
related to poleward or equatorial shifts in westerly winds.
**2.5    Snowpack energy accounting**
Accurate measurement of snowpack energy balance and associated melt can be difficult due to snowpack
heterogeneity (Reba et al., 2009) and problems with energy balance closure (Helgason and Pomeroy, 2012). The
basic snowpack energy balance can be expressed as:
$$Q_m = Q^* + Q_h + Q_e + Q_g + Q_r \tag{1}$$
where the energy available for snow melt ($Q_m$) is equal to the sum of $Q^*$, $Q_h$ and $Q_e$, $Q_g$, and the energy flux to
the snowpack from liquid precipitation ($Q_r$) (Male and Granger, 1981;McKay and Thurtell, 1978). It's important
to note that all terms used in the calculation of the snowpack energy balance are net terms (Marks and Dozier,
1992;Stoy et al., 2018;Welch et al., 2016). Using net terms allows for conservation of energy within the (ideally)
closed energy balance system of the snowpack and aids in more accurately determining contributions of each term
to the energy balance.
Internal energy storage and melt processes can make calculation of the snowpack energy balance particularly
difficult when internal measurements of the snowpack are not available due to problems closing the energy
balance (Helgason and Pomeroy, 2012). This is particularly difficult over Australia's snowpack due to its marginal
characteristics that result in nearly constant internal snowpack melt. Therefore, $Q_m$ can be more accurately
expressed as a residual energy term ($Q_{res}$) that is defined as the sum of the measured terms in Eq. (1) plus any
error in energy balance closure ($Q_{ec}$):
$$Q_{res} = Q^* + Q_h + Q_e + Q_g + Q_r + Q_{ec} \tag{2}$$
While $Q^*$ can be used for basic analysis of the snowpack energy balance, a decomposition into its individual
components is necessary to understand the role of short and longwave radiation exchange in snowpack energetics
(Bilish et al., 2018). Therefore, net radiation should be broken into its net flux terms:
$$Q^* = K^* + L^* \tag{3}$$
that quantify the net shortwave ($K^*$) and net longwave ($L^*$) components.
The approach taken within this paper is to examine net radiative flux components individually, similar to the
methods used by Bilish et al. (2018), to be precise in the identification of synoptic-scale effects on snowpack
energy fluxes through differences in temperature, relative humidity, cloud cover. $Q_{res}$ calculation and
comparisons of snowpack energy flux terms were performed using the terms in Eq. (2), but with the net radiation
terms ($K^*$ and $L^*$) used rather than summed as $Q^*$ only. This research uses the energy flux convention where
positive values are flux to the snowpack and negative values are flux away from the snowpack.





**2.6    Energy flux measurements of synoptic types**
Coordinate rotation for EC systems is typically used to account for errors introduced into flux data due to
imprecise instrumentation levelling. However, complex terrain can complicate EC measurements through local
scale processes such as thermally induced anabatic and katabatic flows, modification and generation of complex
terrain-induced flows, and inhomogeneity of terrain. In these areas, coordinate rotation is used to align the eddy
covariance coordinate system with the sloping surface and to identify and remove larger scale motions that may
be measured with the microscale flows. The Pipers Creek catchment site is located on predominantly level terrain,
however, double coordinate rotation was used to process the EC data to ensure terrain-induced influences on
airflow were removed (Stiperski and Rotach, 2016).
Frequency corrections were made to the EC data to account for sensor response delay, volume averaging, and the
separation distance of the sonic anemometer and gas analyser when calculating fluxes. Finally, WPL air density
corrections (Webb et al., 1980) were made to account for vertical velocities that exist as a result of changing air
mass density through fluxes of heat and water vapour. Quality flags were calculated for $Q_h$ and $Q_e$ using the
methods of Mauder and Foken (2011) that assigned a number from 0-2 based on the quality of the fluxes. High
quality data that is able to be used for fundamental research was assigned a 0, fluxes assigned a 1 are less accurate
but can still be used for long term observations, and fluxes assigned a 2 needed to be removed and gap-filled.
$Q_h$ and $Q_e$ flux were calculated using the EC equations:
$$Q_h = -\rho C_p (\overline{w'\theta'}) \qquad (4)$$
$$Q_e = -\rho L_v (\overline{w'q'}) \qquad (5)$$
where $\rho$ is air density (kg m⁻³), $C_p$ is the specific heat of air (1005 J kg⁻¹ deg⁻¹), $\overline{w'\theta'}$ is the average covariance
between the vertical wind velocity $w$ (ms⁻¹) and potential temperature $\theta$ $(K)$, $L_v$ is the latent heat of sublimation
or vaporization of water (J kg⁻¹), and $\overline{w'q'}$ is the average covariance between the vertical wind velocity $w$ (ms⁻¹)
and specific humidity $q$ (kg kg⁻¹) (Reba et al., 2009).
The calculation of $Q_r$ followed Bilish et al. (2018) and was determined using three separate calculations to
establish approximate wet bulb temperature $(T_w)$ (Stull, 2011), the fraction of precipitation falling as rain $(1 -$
$P_{snow})$ (Michelson, 2004), and total rain heat flux $(Q_r)$ based on precipitation accumulation over a 30-minute
period.
**2.7    Energy flux data quality control and gap-filling**
In addition to removing EC measurements assigned a quality flag of 2, $Q_e$ and $Q_h$ values were also removed
when water vapour signal strength, a unit-less number calculated from the fraction of beam received compared to
that emitted, from the gas analyser was < 0.70 in order to remove erroneous readings during periods of
precipitation (Campbell Scientific, 2018;Gray et al., 2018). A seven point moving-median filter was implemented
over three iterations to de-spike the data and remove values more than 3.0 standard deviations away from the
median values.
Pre-existing gaps and gaps introduced into the data by the quality control procedures were filled using linear
interpolation described by (Falge et al., 2001a;2001b) and the Random Forest regression technique (Breiman,



2001). Linear interpolation of missing $Q_e$ and $Q_h$ values was used for gaps up to 90 minutes in length.
Traditionally, mean diurnal variation values are also used for gap filling procedures (Falge et al.,
2001a;2001b;Bilish et al., 2018). However, it was determined that using mean values would likely obscure any
unique energy balance characteristics of the synoptic types being investigated and, therefore, was not included as
a gap-fill strategy for the data.
The R programming package randomForest (Liaw and Wiener, 2002) was used to fill gaps in $Qe$ and $Qh$ longer
than 90 minutes in length. The random forest regression trained to determine $Q_e$ and $Q_h$ flux values was developed
using twenty-six atmospheric and soil variables collected in addition to EC measurements. Mean squared errors
(MSE)'s were examined for forests with 1-500 trees and it was determined that 150 trees were sufficient to build
an accurate model for both $Q_e$ and $Q_h$. Tests were then conducted to determine the optimal number of variables
to be randomly selected at each node that showed 13 variables was optimal for determination of $Q_h$ and 14
variables should be used for $Q_e$. The $Q_e$ and $Q_h$ random forest regression models were tested for their ability to
predict values that had been used to train the models by comparing the measured $Q_e$ and $Q_h$ values with the
predicted values. Root Mean Squared Error (RMSE) and the Coefficient of Determination ($R^2$) were determined
for each advective flux. Predicted values showed high correlation to measured values with both variables showing
$R^2$ values higher than 0.97. The $Q_e$ regression had a RMSE of 2.56 Wm$^{-2}$ and had lower uncertainty than the $Q_h$
regression that had a RMSE of 4.67 Wm$^{-2}$.
Following quality control procedures, 2571 of the initial 7756 records (33%) remained in the $Q_e$ data and 4019
records (52%) remained in the $Q_h$ data. Linear interpolation yielded an addition of 910 $Q_e$ values (12%) and 928
$Q_h$ values (12%). The Random Forest regression models were the largest source of gap-filled data with the
contribution of an additional 4275 $Q_e$ values (55%) and 2809 $Q_h$ values (36%).
## 3    Results
Identification of homogeneous snow cover days for the 2016 and 2017 snow seasons (June to October) resulted
in 163 total days with 90 days occurring in the 2016 and 73 days in 2017. July, August, and September had the
highest number of classifiable days during the period. June and October still had periods with homogenous snow
cover, but they became intermittent and fewer classifiable days were in each of the months. This led to fewer
periods of study at the beginning and end of the snow seasons when the snowpack was variable, with more in the
late winter and early spring months when snow cover was more consistent. Mean surface and cloud characteristics
and median daily energy flux characteristics of synoptic types identified during the two seasons are presented in
Table 2.
### 3.1    Synoptic types
#### 3.1.1    Surface characteristics
The dominance of the subtropical ridge in Australia's mid-latitudes is evident in the synoptic types. Four of the
types (T1,T2,T5 and T7) display dominant surface high pressure systems, each with slightly different orientation
and pressure centre locations (Figure 3a) resulting in different energy flux characteristics. Dominant south-
southwesterly winds from T1 are the result of the high pressure centre being located to the northwest of the study
area. T2 has a predominantly zonal flow resulting from an elongated high to the north-northeast. T5 and T7 are





characterized by north-northwesterly flow from high pressure centres over the New South Wales
(NSW)/Queensland (QLD) coast and directly over the Snowy Mountains region, respectively.
T3 is characterized as having dominant northwest winds along a trough axis that is positioned over SEA with a
secondary coastal trough extending from southern NSW to the NSW/QLD border. T4 shows a transition from a
surface trough that has moved to the east of the study region to a high pressure system that is moving into the area
with winds from both features that converge over the Snowy Mountains region. The only synoptic type to have
dominant influence from a surface low was T6 that had weak south-southwesterly flow over the region from a
weak cut-off low to the east. For the purposes of this research, the identification of cut-off lows follows the
characteristics outlined by Chubb et al. (2011) that omits the presence of a closed circulation, but includes a cold
anomaly aloft that was cut off from the westerly wind belt.
Though characterization of synoptic types is purely statistical, T1, T4, T5, and T6 are considered to be 'transition
types' as they have surface pressure characteristics that indicate a change in pressure regime (low – high or high
– low) in the upcoming days. T1, T4, and T6 are post-frontal transition types that show high pressure ridging into
the region following the passage of a trough that has either moved to the east (T1 and T4) or developed into a
weak lee-side cut-off low (T6). T5 shows the approach of a trough from the west and an associated transition to
a low pressure system. T2 and T7 show the area under the influence of zonal flow as a result of high pressure
systems centred over the area, while T3 shows SEA under the influence of a trough at the time of observations.

### 3.1.2    Relative humidity and cloud cover

Understanding RH values associated with different synoptic types provides the ability to track types that are
favourable for high $Q_e$ exchange with the snowpack. In addition, RH values at all tropospheric levels can have
impacts on snowpack energy flux through influences on $K^*$ and $L^*$ exchange via changes to insolation and the
absorption and emission of $L$. The identification of RH characteristics and associated cloud cover is necessary to
fully develop energy flux characteristics for each type.
Many of the synoptic types display local RH maxima in the Snowy Mountains region at 850 hPa (Figure 3b) and,
while T5 has the lowest RH values of all types, it still has slightly higher RH values over the area. The elevation
in RH values in the region is most likely caused by changes of airmass thermodynamic properties due to
orographic forcing of the mountains (Ahrens, 2012). T4 and T6 had the highest RH values over the region at 850
hPa with both being widespread and higher than 90%. T6 shows strong southerly advection of elevated RH values
from the tropics along the NSW and QLD coast ahead of troughs at 700 and 500 hPa that are associated with the
surface cut-off low.
Identification of cloud cover, conducted following the procedures outlined in section 2.4, agreed with the mean
RH characteristics of T4 and T6 with both types having 100% cloud cover between partial and complete cloud
cover days (Table 2). T6 showed the highest RH values of any type with values greater than 90% over the region
at the 700 and 500 hPa levels. While not definitive, this would suggest that T6 has deeper or more cloud layers
than T4, which likely only has clouds at lower altitudes. T2 and T7 had the lowest percentage of days with any
cloud cover, which is confirmed by their low RH values at 700 hPa (<20% & <30%) and 500 hPa (<30% &
<40%), respectively. In addition, they also had the highest percentage of cloud-free days with T2 clear sky 19%



of the time and T7 having 23% of its days without cloud. The remaining types (T1, T3, and T5) showed a relatively
consistent number of cloud days based on the satellite observations that were all above 85%.

### 3.1.3    Temperature

Temperature characteristics of synoptic types at low and mid-levels in the atmosphere are crucial to identify those
with the highest surface sensible heat flux characteristics. The highest mean temperatures and strongest warm air
advection (WAA) in the Snowy Mountains region at 850 hPa (Figure 3c) was found to be from T5 that is driven
by converging winds on the back of a high pressure circulation to the east and the leading edge of a trough to the
west. T2 and T3 have the second and third highest temperatures, respectively, but have different advection
characteristics. T2 shows relatively weak WAA into the Snowy Mountains region associated with zonal flows at
850 hPa resulting from the high pressure circulations located to the north (similar to T7). However, T3 shows cold
air advection (CAA) associated with dominant winds from the west-northwest.
Overall, CAA at 850 hPa can be identified in four of the seven types (T1, T3, T4, and T6) and warm air advection
exists in the other three synoptic types (T2, T5, and T7). Of the four CAA types, T1 and T4 advection is being
generated through south-southwest and west-southwest winds, respectively, related to high pressure centres to the
northwest. Despite a stronger southerly component of dominant CAA winds in T1, temperatures are lower in T4
which has a higher westerly component to the wind. T6 shows CAA related to converging winds on the back of a
trough to the east and a high to the northwest.

### 3.1.4    Frequency and duration

The frequency of each synoptic type during the 2016 and 2017 snowpack seasons is shown in Table 2. T3 and T7
occurred most frequently with 26.99% (44 days) and 19.02% (31 days) respectively. The higher number of days
in T3 and T7 is reflected in the mean type duration that shows these types with the longest duration. This is likely
due to these synoptic types occurring in a slower progressing synoptic pattern over multiple days as seen in the
mean type duration data (Table 2).
Transition probabilities for the 2016 and 2017 seasons were developed similar to those used by Kidson (2000)
that detail the likelihood of a synoptic type occurring on the following day given an initial type. The highest
transition probabilities were identified for each type and a flowchart was developed based on the most likely
synoptic type progressions (Figure 4a). If the highest transition probabilities were within < 0.05 of each other, two
paths were plotted. The flowchart shows what would be expected for a basic synoptic-scale circulation at mid-
latitudes; a trough propagating eastward into the Snowy Mountains region in T7, T5, and T3; either continued
eastward movement of the surface trough (T4) or the development of a weak cut-off low (T6); then transitioning
to dominant high pressure over the region again (T2, T1, or T7).

### 3.2    Energy flux characteristics of synoptic types

It is important to consider the effects of synoptic type frequency when determining primary sources of energy
fluxes over long periods, as synoptic types that contribute the most to snowpack ablation may simply have a higher
rate of occurrence and lower daily energy flux values than other types. In order to obtain a more detailed
understanding of each type's energy flux, median daily energy flux calculated for each type was determined to be
a better method of comparison. Therefore, both median daily and total snowpack fluxes over the two seasons
(Figures 5 & 6) are presented in MJ m$^{-2}$ to show synoptic type energy flux contributions made at short and longer





temporal scales. While initial measurements were made in Wm$^{-2}$, the use of MJ m$^{-2}$ in this paper allows for easier
comparison to other energy balance works conducted on this region (Bilish et al., 2018) as well as research on
synoptic weather and energy fluxes in other locations (Welch et al., 2016;Burles and Boon, 2011;Ellis et al.,
2011;Hay and Fitzharris, 1988;McGregor and Gellatly, 1996;Granger and Gray, 1990;Neale and Fitzharris, 1997).

### 3.2.1    Latent and sensible heat flux

Daily $Q_e$ was negative for each of the seven synoptic types (Figure 5a) and the magnitude of the values was shown
to correspond to the mean 850 hPa RH values for each type reflecting the site elevation of 1828 m asl. Two of the
three types with the lowest RH values (T2 and T5) showed the greatest negative $Q_e$ values and those with the
higher RH values (T1 and T6) showed the least amount of $Q_e$, which is consistent with conditions needed for
evaporation from the snowpack. T5 had the second largest negative $Q_e$ values of any type with a median value
of -1.00 MJ m$^{-2}$ day$^{-1}$ which corresponds to its low 850 hPa RH values, the highest observed surface mean daily
ambient temperature of 3.5 °C, and the second lowest observed surface mean RH value of 65% with only T2 being
lower (60%). T3 showed the largest release of $Q_e$ from the snowpack with a median value of -1.11 MJ m$^{-2}$ day$^{-1}$.
Overall, negative $Q_e$ was offset by positive $Q_h$ for most synoptic types with the exception of T3 that had mean
surface temperatures below zero (-0.83°C) and a measured surface RH value below 90% resulting in more $Q_e$
loss than $Q_h$ gain by the snowpack. Similar to trends seen in $Q_e$, the highest daily median $Q_h$ values (Figure 5b)
were associated with synoptic types with the highest temperatures at 850 hPa (T5, T7, & T2), which coincided
with observed temperatures from the energy flux tower (3.48°C, 1.46°C, & 1.89°C). T5 showed the highest daily
$Q_h$ values as a result of having the highest temperatures and also has the second lowest $Q_e$ value that is associated
with having the lowest RH of any type (60%). Ultimately, when both turbulent terms are considered, T5 had the
highest amount of energy flux into the snowpack (1.49 MJ m$^{-2}$ day$^{-1}$) followed by T7 (1.40 MJ m$^{-2}$ day$^{-1}$) and T1
(1.00 MJ m$^{-2}$ day$^{-1}$).

### 3.2.2    Radiation flux

The largest contribution of radiative energy to the snowpack from all synoptic types was $K^*$ which accounted for
53-97% of total positive flux (Figure 5c). By comparison, $L^*$ accounted for 61-95% of negative energy flux from
the snowpack (Figure 5d) with the highest amounts of loss belonging to the types with the lowest percentage of
cloud cover (T1, T2, and T7). Total radiation flux varied largely by synoptic type and was found to be positive in
types T3 and T6 and negative for the rest of the types. The two types with positive net radiation had the highest
incoming longwave radiation flux values mostly balancing outgoing longwave values. This meant that incoming
shortwave radiation to dominates Q*. The largest loss in Q was exhibited by T1, that was 31% higher than the
next closest type (T4).  The types with net radiation loss (T1, T2, T4, T5, and T7) had values that ranged from -
0.67 MJ m$^{-2}$ day$^{-1}$ (T5) to -2.78 MJ m$^{-2}$ day$^{-1}$ (T1). However, T4 had dissimilar cloud and RH characteristics to
T2 and T7, which had the two lowest cloud cover percentages and two of the lowest RH values. T4 had 100%
cloud cover and had an associated reduction in incoming shortwave radiation that allowed the outgoing longwave
radiation term to become more dominant than in T2 or T7 and, therefore, gave it the highest Q* loss of the three.

### 3.2.3    Ground and precipitation heat flux

Energy flux from ground and $Q_r$ (Figure 5e & 5f) were the smallest of any term for all synoptic types, with $Q_g$
and $Q_r$ accounting for less than one percent of median daily energy fluxes for all synoptic types. Ground heat flux



characteristics were similar between all synoptic types and varied little. While $Q_r$ was small when examined as a
daily median value, it does show a high degree of variation primarily associated with T5 and T3. This is due to
several large rain events that occurred during 2016 (18 July; 21 and 22 July; and 31 August) and one during 2017
(15 August). Despite relatively low energy flux contributions by rainfall, it is interesting to note that the ten days
with the highest rainfall fluxes (>0.05 MJ m$^{-2}$ day$^{-1}$) consisted of four T5 days, three T3 days, two T7 days, and
one T6 day showing a significant clustering of high precipitation days in types T5 and T3.

### 3.2.4    Total daily net energy flux

Overall, two synoptic types (T5 and T6) had positive median daily net energy flux to the snowpack (Figure 6a).
Of these, T5 had the largest energy flux that was related to its relatively high temperatures that contributed to the
highest $Q_h$ value of any synoptic type and increased solar radiation from less cloud cover. Contrary to the reduction
in cloud cover that aided T5 in having the highest total energy flux contributions, T6 had the highest cloud cover
and yet had the second highest energy flux to the snowpack that was primarily due to increased incoming
longwave radiation. T7 was close to having neutral energy fluxes with a median value of only -0.04 MJ m$^{-2}$ day$^{-1}$
as a result of relatively low percentage of cloud cover resulting in strongly negative $L^*$ as well as the second
highest $Q_h$ term of any type.
T1 and T4 showed the greatest negative median daily net energy flux of all synoptic types (Figure 6a), which
could be attributed to their negative $L^*$ and to having low $K^*$ terms. T3 has a similar net energy flux to T4, but is
negative primarily due to having the only negative $Q_h$ of any type. T2 also had a net negative median daily energy
flux but to a lesser extent than either T1, T3, or T4. Relative humidity values lower than any other type were the
primary driver behind T2's negative net value as it resulted in the highest longwave radiation loss from the
snowpack through having the lowest cloud cover, as well as $Q_e$ loss.
The synoptic type T5 contributed the most energy to the snowpack during the two seasons (Figure 6b) due to a
high number of occurrences (24) and having the largest positive fluxes from high $Q_h$ values associated with strong
WAA ahead of the passage of cold fronts. While T6 was the only other type to have positive median daily energy
flux contributions to snowpack energy flux, T7 contributed a higher amount of energy flux during the two winter
periods because it had the second highest number of occurrences, and the distribution of occurrences around the
median show that events were either near-neutral or positive in their energy fluxes. T6 was the only other type to
have a positive energy flux contribution to the snowpack over the two seasons and it was smaller than that of T5
or T7. Similar magnitude was seen in the negative flux contributions of T1, T2, and T4 with T2 having the most
significant negative flux. T1 and T4 also showed negative fluxes, but T3 showed a nearly neutral contribution to
snowpack energy flux over the two winter seasons. As T3 is associated with a surface trough, it's possible that
pre-frontal and post-frontal characteristics are both incorporated in the energy balance of T3 and act to cancel
each other out when averaged over a longer period.
All synoptic types had variation in median daily net energy that can be attributed to the classification conducted
by the k-means clustering technique. Each type consisted of classified days that had similar synoptic
characteristics, but differences in system strength and position affected energy fluxes for individual days.
Therefore, it is important to remember that each synoptic type is associated with a range of daily energy flux
values in addition to the median daily energy flux for each type.



## 4 Discussion

### 4.1 Properties of synoptic type energy balance

Net shortwave radiation flux contributed the largest amount of energy to the snowpack for all synoptic types ranging from 53-97% of median daily energy flux with T5 being the only synoptic type below 60% contribution (53%) of $K^*$ to the snowpack. These results agree with Fayad et al. (2017) who noted that radiative fluxes are the dominant source of snowpack melt energy in mountain ranges with Mediterranean climates. Net $Q_h$ contributed the second highest percentage of median daily energy flux to the snowpack accounting for 16-44% of positive fluxes with the exception of T3 that had a $Q_h$ term that accounted for 4% of its negative fluxes. The largest contributions of $Q_h$ to the snowpack are associated with synoptic types T2, T4, T5, and T7 that are characterised by high pressure and northwesterly or westerly winds that are associated with WAA. Hay and Fitzharris (1988) noted that, while radiative terms were responsible for the majority of energy contributions to glacier melt in New Zealand's Southern Alps, turbulent fluxes contributed significant amounts of energy to melt. Similarly, despite $Q_h$ not being the dominant energy flux to the snowpack for any synoptic type, it does account for nearly half of the energy flux to the snowpack for T5 (44%) and over a third for T7 (35%), and is still a significant source of energy flux to the snowpack for nearly all synoptic types.

Median daily energy loss from the snowpack was from $Q_e$ and $Q^*$, which dominated T1, T2, and T4 resulting in negative median daily energy fluxes from the snowpack. Net longwave radiation was the most influential term in the emission of energy from the snowpack accounting for 61-95% of energy loss with net $Q_e$ flux accounting for 5-39% of outgoing energy flux. Though the methodology of this paper distinguishes between shortwave and longwave fluxes in order to better examine the effects of synoptic-scale features such as RH or cloud cover on radiative transfers similar to that of more recent works such as Cullen and Conway (2015), many historical works have not made the same distinction in terms (Moore and Owens, 1984;Hay and Fitzharris, 1988;Neale and Fitzharris, 1997;Stoy et al., 2018). It should be noted that had $Q^*$ been used for comparison, the results of this paper agree with several studies (Sade et al., 2014;Moore and Owens, 1984;Bednorz, 2008b) that found that turbulent fluxes were the dominant fluxes when examining the energy flux characteristics on snowpacks in climates similar to that of the Snowy Mountains in the Australia Alps.

Median daily $Q_g$ values were found to account for only a small fraction of total energy flux to the snowpack consisting of 1-5% of daily positive energy fluxes. Similarly, energy flux to the snowpack from $Q_r$ has been shown to only contribute < 1% of total seasonal energy flux for five of the seven synoptic types which agrees with the findings of other studies (Bilish et al., 2018;Mazurkiewicz et al., 2008). However, precipitation was responsible for > 1% of the daily median energy flux of the two synoptic types primarily associated with rain-on-snow events, T5 and T3. Although fluxes imparted on the snowpack from rainfall are relatively small when compared to all positive fluxes, the accompanying energy flux characteristics of T5 associated with rain-on-snow events are responsible for two of the three largest contributions of overall snowpack energy fluxes.

The results show a significant agreement with previous research conducted in this region by Bilish et al. (2018) when methods from that work are used to calculate relative contributions of positive energy fluxes to the snowpack. Overall, incoming longwave radiation was shown to be the highest positive flux to the snowpack accounting for 75-86% of incoming energy flux. Shortwave radiation was responsible for an additional 8-14% of





incoming energy flux with $Q_h$ accounting for 0-9% of incoming fluxes, $Q_e$ generating 0-4%, $Q_g$ attributing 0.3%,
and $Q_r$ accounting for 0.1%. Despite methodological differences that can be attributed to the need to highlight
different processes within atmosphere – snowpack interaction, results from both papers show similar overall
energy fluxes.

**4.2    Synoptic patterns and energy flux**

Snowpack energy flux characteristics recorded at the Pipers Creek catchment headwaters have been related to
synoptic weather types that occurred during the 2016 and 2017 snow seasons. The resulting analysis reveals a
maximum in positive energy flux as pre-frontal troughs approach the Snowy Mountains, followed by cold front
conditions during the T7→T5→T3 common progression pattern identified here. Several factors cause high
positive energy flux during these periods that include: an increase in temperatures due to WAA and the associated
increase in positive $Q_h$; decrease in negative $L^*$ due to an increase in cloud cover; a decrease in $Q_e$ following
frontal passage and associated increase in RH; and progressively increasing $Q_r$ as the trough approaches and
immediately after passage.
Synoptic types characterized by surface high pressure as their primary influence (T1, T2, T4, and T7) had four of
the five negative daily contributions to snowpack energy flux. In T1, T2, and T7, net shortwave radiation terms
($K^*$) were positive and varied by ~4-10% for these types, however, low RH and cloud cover allowed for highly
negative $L^*$ terms that were not compensated by change in $K^*$. In contrast, T4 had higher cloud cover and increased
RH that were due to advection of moisture from the Tasman Sea. The higher RH in T4 and low mean air
temperature (-2.06°C) resulted in $Q_e$ and $Q_h$ terms of similar magnitudes, but opposite signs that nearly cancelled
out. This resulted in a $L^*$ term that was of lesser magnitude than those of T1, T2, and T7, but still the dominant
term in its energy exchange.
Four primary synoptic circulation patterns were identified during the study period. Each of the four patterns and
their associated energy flux values calculated from median daily flux and mean type duration can be seen in
Figures 4a and 4b. While each pattern differs towards the end of the cycle, each one has the T7→T5→T3
progression in common. Unsurprisingly, the highest contribution of median energy flux to the snowpack (0.75 MJ
m$^{-2}$) is from Pattern 1, which has only one synoptic type with negative flux (T3) whereas the others all contain
multiple negative flux types. Pattern 3 had the largest negative snowpack energy flux (-2.44 MJ m$^{-2}$) due to it
containing types with the highest net energy loss (T1 and T4).
Changing synoptic regimes in the Snowy Mountains suggest an increase in anti-cyclonic conditions (Hendon et
al., 2007), such as types T1, T2, T4, and T7, as a result of poleward shift in the subtropical ridge (Cai et al., 2005).
Under these conditions, snowpack energy exchange in the Australian Alps would be expected to decrease as
synoptic types related to anti-cyclonic conditions have negative energy fluxes to the snowpack and synoptic
patterns T3 and T4, which have the largest negative snowpack energy fluxes, would increase in frequency. While
these results may seem counterintuitive regarding a generally warming climate, they agree with the findings of
Theobald et al. (2016) who showed reductions in cool-season precipitation amounts and frequency due, in part,
to reductions in the occurrence of dominant cold front systems. The reduction in cold-frontal systems in the
Australian Alps region is associated with declines in the pre-frontal WAA that has been shown to be the primary
driver of positive snowpack energy flux. However, potential reductions in energy fluxes to the snowpack will not





likely lead to increases in snowpack duration or depth, as reductions in precipitation are associated with the shifts
to anti-cyclonic synoptic patterns (Theobald et al., 2016;Theobald et al., 2015).
**4.3    Distribution of gap-filled eddy covariance fluxes**
One of the disadvantages of the Random Forest regression method to gap-fill missing EC data is that exact results
aren't reproducible due to the method's random handling and sub-setting of predictor variables. Methods of
developing models and predicting values were evaluated over twenty iterations to determine the amount of
variability in RMSE when generating a random forest from the same dataset. Some variability in RMSE was noted
between tests for $Q_e$ and $Q_h$ but was small with a standard deviation of 0.01 Wm$^{-2}$ in $Q_e$ and 0.03 Wm$^{-2}$ in $Q_h$.
Small differences in RMSEs between model development runs and data filling indicate that RMSE values for gap-
filled data would be best represented as 2.56 ± 0.01 Wm$^{-2}$ for $Q_e$ and 4.67 ± 0.03 Wm$^{-2}$ for $Q_h$
Gap-filling of $Q_h$ and $Q_e$ can introduce uncertainty into measurements that may affect the ability to thoroughly
compare datasets such as those pertaining to the different synoptic types compared within this work. As such, it
is important to note that not all synoptic types had equal amounts of gap-filling for their $Q_e$ and $Q_h$ fluxes.
Distribution of gap-filled data within synoptic types depended largely on the quantity of precipitation associated
with each type. The most significant concentrations of gap-filled data were in T3 ($Q_e$: 74%, $Q_h$: 55%) T5 ($Q_e$:
57%, $Q_h$: 39%), and T6 ($Q_e$: 81%, $Q_h$: 73%). Differences in the quantity of gap-filled data between synoptic types
can create uncertainty when making comparisons between fluxes in each. However, uncertainty introduced
through gap-filling procedures is relatively low and should have a minimal impact during comparison of fluxes.
**5    Conclusions**
Overall, periods of pre-cold frontal passage contribute the most energy fluxes to snowpack melt due to WAA
ahead of the front, a reduction in cloud cover allowing for higher incoming shortwave radiation, and the gradual
development of precipitation that often contributes to rain-on-snow events. While this work was conducted solely
on the Australian snowpack, snowpacks in other regions such as New Zealand (Hay and Fitzharris, 1988;Neale
and Fitzharris, 1997), Canada (Romolo et al., 2006a;2006b), the Spanish Pyrenees (Lopez-Moreno and Vicente-
Serrano, 2007), and the Arctic (Drobot and Anderson, 2001) see similar synoptic-scale effects on snowpack
energy to those presented here. Snowpack energy fluxes in the Australian Alps would likely decrease under
climate change progression as a result of reductions to primary cold-frontal systems and associated pre-frontal
WAA.
The understanding of synoptic-scale processes on snowpack energy balances will likely become applicable to
broader regions as climate change continues and snowpacks develop warmer properties (Stewart, 2009;Adam et
al., 2009). An increased burden on freshwater systems for agriculture, drinking water, and energy production will
continue as these changes occur (Parry et al., 2007). Therefore, continued work on marginal snowpack ablation
processes, such as those within the forested regions of Australia's Snowy Mountains, will be important to resource
management and should be explored.
**Data Availability**
Energy flux data used in this study is available at https://doi.org/10.14264/uql.2019.691. ERA-Interim reanalysis
data are freely available from the European Centre for Medium-Range Weather Forecasts
(https://www.ecmwf.int/en/forecasts/datasets/reanalysis-datasets/era-interim). Precipitation data used in this



study was supplied by Snowy Hydro Limited via restricted access, this data can be obtained by contacting Snowy
Hydro Ltd.
**Author Contributions**
AS, HM, AT, and NC designed the experiments and AS conducted them. AT developed the k-means clustering
and synoptic typing code. AS developed the code related to energy balance and eddy covariance measurements.
AS wrote the manuscript with input from all authors.
**Competing Interests**
The authors declare that they have no competing interests.
**Acknowledgements**
The authors would like to thank Shane Bilish for establishment of the Pipers Creek snowpack research catchment,
Michael Gray for installation and maintenance of the energy balance tower, and the Weather and Water team at
Snowy Hydro Limited for their contributions of data and field support during the data collection and analysis
process. AS was supported by an Australian Government Research Training Program Scholarship.






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











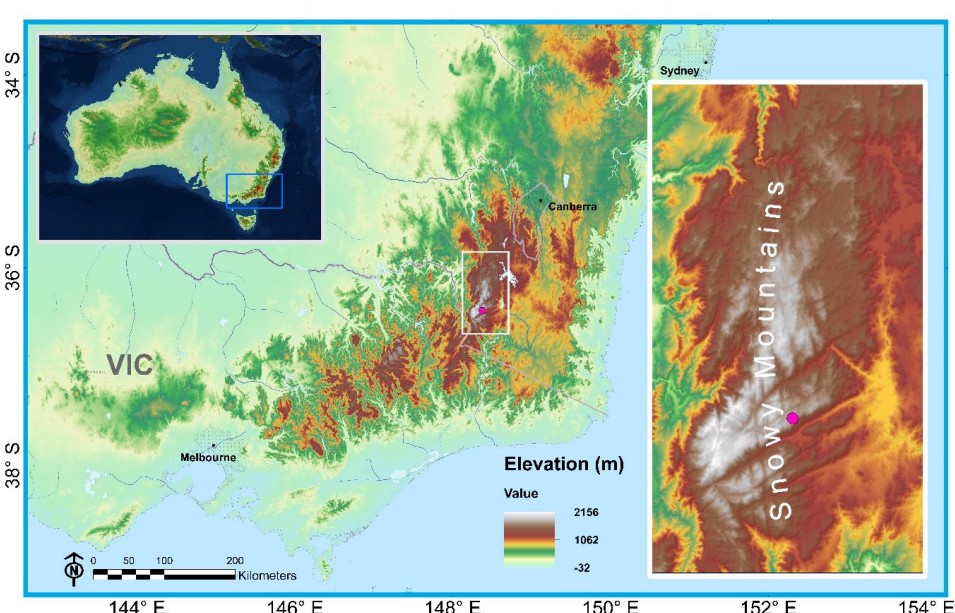

**Figure 1: Map of southeast Australia and the Snowy Mountains. Pink dot represents the location of the energy balance instrumentation site. Map layer sources copyright: ESRI, USGS, NOAA, DigitalGlobe, GeoEye, Earthstar Geographics, CNE S/A Airbus DS, USDA, AeroGRID, IGN, and the GIS User Community.**















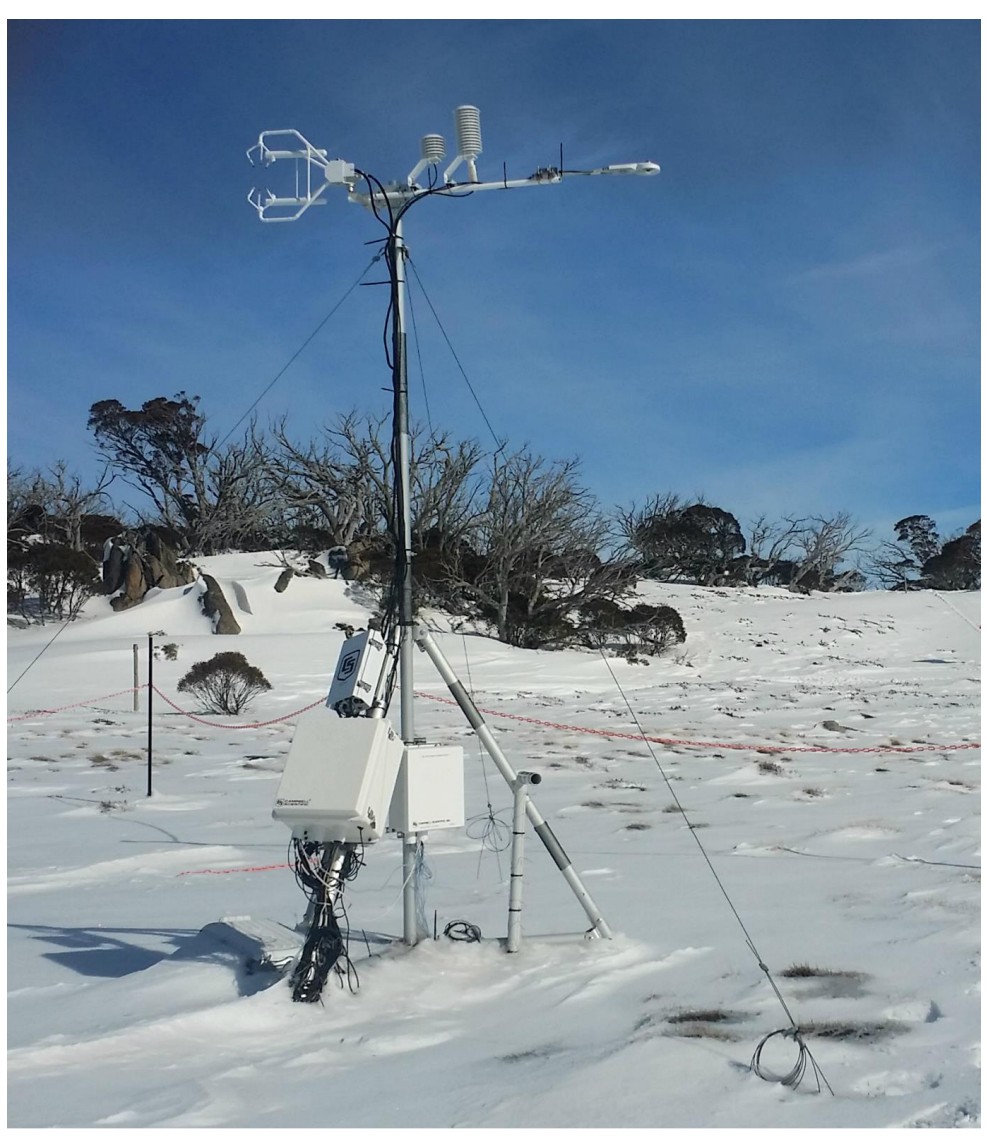



**Figure 2: Energy balance field site with eddy covariance instrumentation at Pipers Creek catchment headwaters.**







**Figure 3: Mean synoptic type MSLP and 10m wind vectors (a), 850 hPa RH and wind vectors (b), and 850 hPa $T_d$ and**
**wind vectors (c) over the southeast Australia region for the 2016 and 2017 seasons. Location of surface energy balance**
**site marked with '+'.**




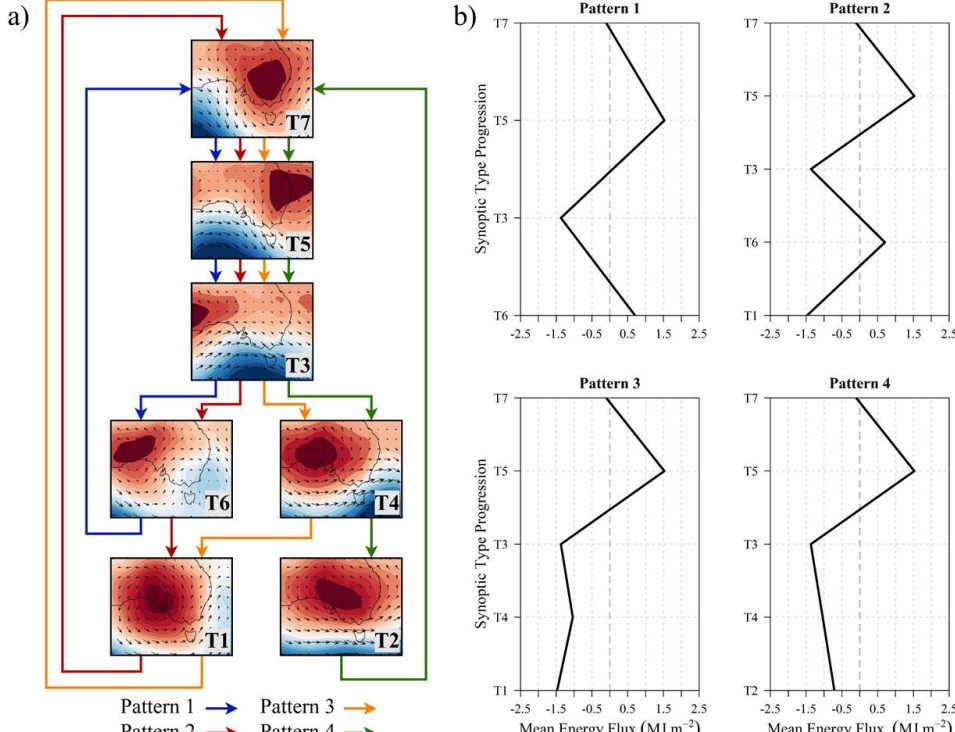


**Figure 4: Flowchart of four primary synoptic type patterns/progressions based on probability of transition for the 2016 and 2017 seasons (a) and calculated synoptic pattern snowpack fluxes based on median daily values and mean duration of synoptic type (b).**










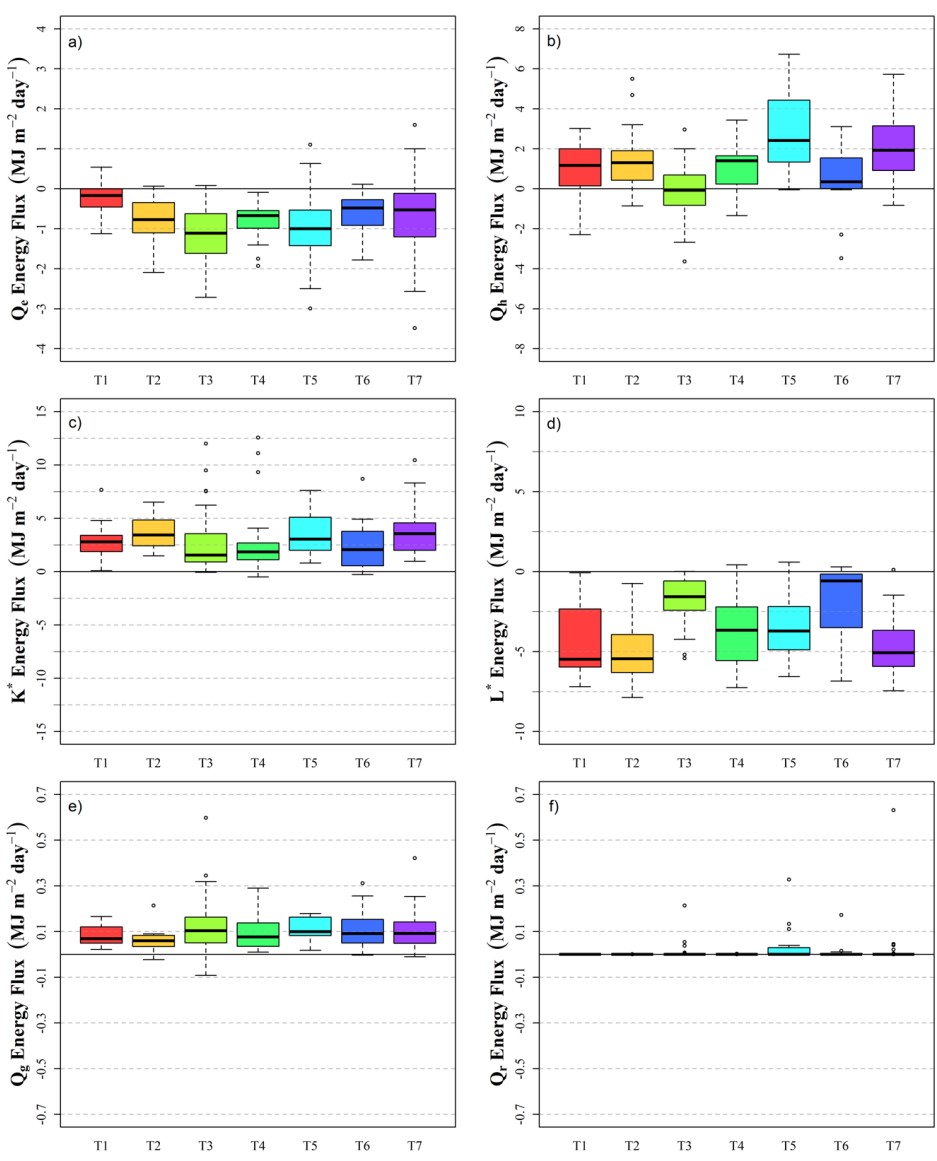

**Figure 5: Daily snowpack energy fluxes by term for each synoptic type for the 2016 and 2017 seasons.**





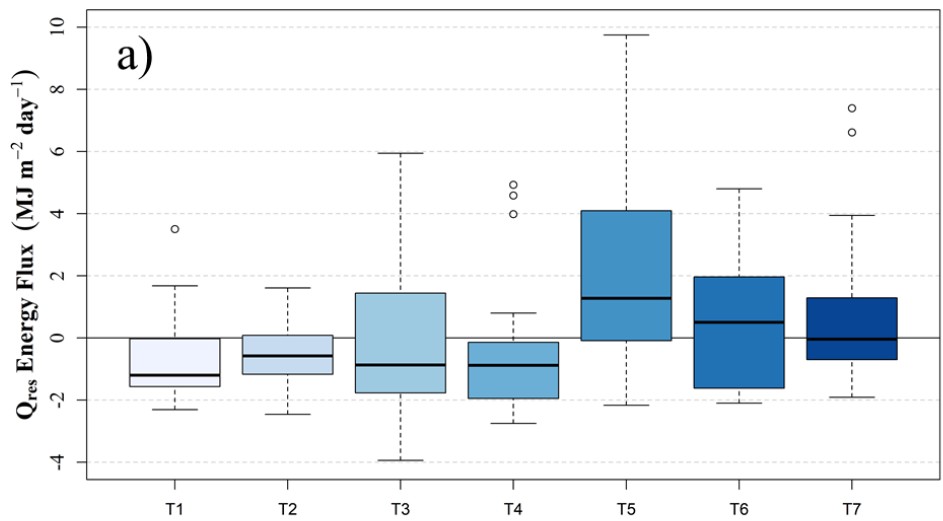

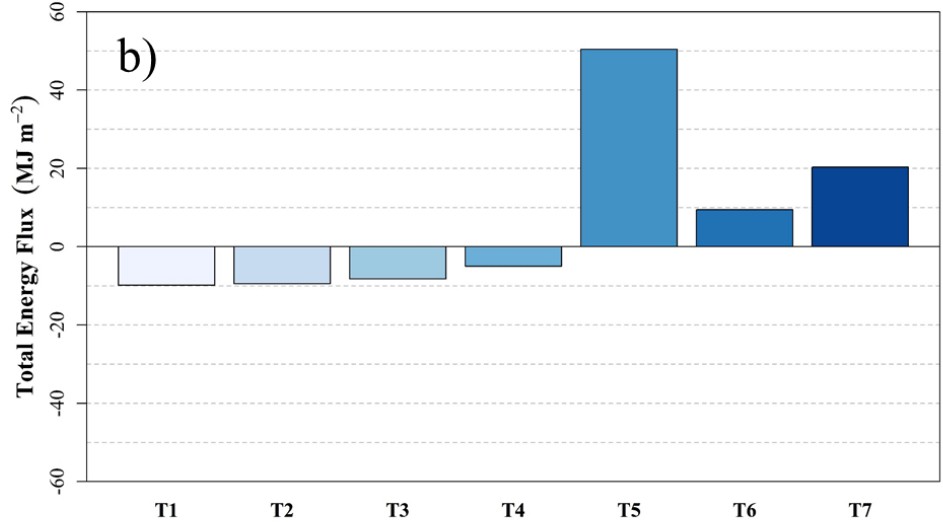


**Figure 6: Daily snowpack energy fluxes (a) and total energy flux (b) by synoptic type for the 2016 and 2017 seasons.**





| Instrument | Manufacturer | Variables Measured | Accuracy |
|---|---|---|---|
| SI-111 | Apogee Instruments | Surface Temperature ($T_{sfc}$) | ± 0.2°C -10°C<T<65°C ± 0.5°C -40°C<T<70°C |
| CS650 | Campbell Scientific | Soil Water Content (SWC) | ± 3% SWC |
|  |  | Soil Temperature | ± 5°C |
| CSAT3A | Campbell Scientific | Wind Components ($u_x$, $u_y$, $u_z$); Wind Speed (u) and Direction (°); and Sonic Temperature | ± 5 cm s$^{-1}$ |
| EC150 | Campbell Scientific | $H_2O$ Gas Density | 2% |
| NOAH II | ETI Instrument Systems | Precipitation Accumulation | ± 0.254 mm |
| HFP01 | Hukseflux | Soil Heat Flux | < 3% |
| CNR4 | Kipp and Zonen | K↓, K↑, L↓, L↑ | K < 5% Daily Total L < 10% Daily Total |
| HMP155 | Vaisala | Air Temperature ($T_d$) | < 0.3°C |
|  |  | Relative Humidity (RH) | <1.8% RH |
| PTB110 | Vaisala | Barometric Pressure | ± 0.15 kPa |


**Table 1: Information on instruments used at the Pipers Creek catchment site.**
















| Synoptic Type | T1 | T2 | T3 | T4 | T5 | T6 | T7 |
|---|---|---|---|---|---|---|---|
| Surface Characteristics | High pressure; SW winds | High pressure; WNW winds | Frontal; NW winds | High/low transition; W winds | High Pressure; NNW winds | Lee-side low; SW winds | High pressure; WNW winds |
| Cloud Cover (% days with any cover) | 87.50% | 76.47% | 89.13% | 100.00% | 87.50% | 100.00% | 76.47% |
| $Q_h$ (MJ m$^{-2}$ day$^{-1}$) | 1.17 | 1.30 | 0.04 | 0.88 | 2.50 | 0.47 | 1.92 |
| $Q_e$ (MJ m$^{-2}$ day$^{-1}$) | -0.22 | -0.64 | -1.16 | -0.67 | -1.09 | -0.51 | -0.53 |
| K↓ (MJ m$^{-2}$ day$^{-1}$) | 12.62 | 15.47 | 8.91 | 11.29 | 12.60 | 8.11 | 13.05 |
| K↑ (MJ m$^{-2}$ day$^{-1}$) | -9.61 | -11.26 | -6.97 | -9.55 | -9.48 | -5.85 | -9.60 |
| L↓ (MJ m$^{-2}$ day$^{-1}$) | 19.53 | 20.16 | 24.95 | 22.08 | 23.59 | 26.57 | 21.38 |
| L↑ (MJ m$^{-2}$ day$^{-1}$) | -25.32 | -26.00 | -26.63 | -25.74 | -27.38 | -26.91 | -26.70 |
| $Q_g$ (MJ m$^{-2}$ day$^{-1}$) | 0.07 | 0.06 | 0.10 | 0.08 | 0.10 | 0.09 | 0.09 |
| $Q_r$ (MJ m$^{-2}$ day$^{-1}$) | 0.00 | 0.00 | 0.00 | 0.00 | 0.01 | 0.00 | 0.00 |
| $Q_m$ (MJ m$^{-2}$ day$^{-1}$) | -1.31 | -0.43 | -0.84 | -0.90 | 1.11 | 0.63 | -0.20 |
| Total Number of Occurrences | 15 | 16 | 44 | 19 | 22 | 16 | 31 |
| Mean Type Duration (Days) | 1.23 | 1.31 | 1.59 | 1.19 | 1.20 | 1.33 | 1.42 |


**Table 2: Synoptic, energy flux, and occurrence characteristics for each synoptic type. Mean Daily surface and cloud**
**cover characteristics are mean values and daily energy flux values are median values.**