# Peer review of "Quantifying the impact of synoptic weather types and patterns on energy fluxes of a marginal snowpack"

_The Cryosphere, 2020_

## Referee Comment (RC1) · Anonymous Referee #1 · 6 Mar 2020

General considerations

This is a resubmission of a paper with a similar title and focus, for which I have provided a quite detailed review on the previous version. The authors have made quite some changes (improvements!) with respect to the first version (data post processing and gap filling), but in my view have not been able to appropriately address the truly major concerns: the representativeness of one surface energy balance (EB) site and the treatment of EB (non-)closure. The authors have decided to 'avoid' the problem (e.g., by no more showing/discussing the evidence with respect to EB closure) or to 'downplay' it (the representativeness issue). I have two major comments making my

points below. My third major concern on the first version had been the identification of synoptic patterns on a daily basis: here, the authors have added two sentences defending their choice – which is fine in principle. However, I still think that the authors miss out some potential or, in other words, introduce some unnecessary variability by choosing a not optimal reference time scale.

Major comments

1) Energy balance closure

The authors have added an additional version of the surface energy balance equation, which, at least formally, addresses the non-closure and introduces the residual (Qres).

→ On l. 221, the authors claim that 'Qres calculation and comparisons of snow pack energy flux terms were performed using the terms in eq. (2)'. This equation contains a 'energy balance closure term' (Qec). This term, however, is not available from the measurements. How did the authors make those 'calculation an comparisons'? (note that the non-closure is not just the sum of the 5 measured terms – because it also includes the Qres (i.e., the energy available for melt and internal [in the snow pack] energy storage).

→ Furthermore, when presenting the results, the 'ec-term' is not shown (and therefore not discussed) – of course, this is no wonder when it was not measured and cannot be derived from the measured terms. What is presented in the results section is the 'total net energy flux' (Section 3.2.4) – but it is not mentioned how this was determined: sum of the 5 measured (Q*, Qe, Qh, Qg, Qr) as in eq (1) [and called Qm]? At least, when comparing Fig. 6a and Table 2 (the entry for Qm), one gets the impression that it is indeed Qm what is now called 'the total energy flux'.

→ finally, in Table 2, Qm is listed, even if on l. 210 it is stated that Qm can be more accurately expressed as Qres. . .

So, overall it appears that the authors have, basically, added a new equation (which is

never used thereafter), do not discuss the issue, and still present the same data - and now seem to call it 'the total energy flux' instead of 'melt energy'. It is, unfortunately, so that the residual is also 'energy flux' – simply not accounted for in the form of the terms in eq (1) [it is local advection, flux divergence, storage, . . .]. This is not a subtlety. In the first version the authors had a short discussion on the energy balance closure (some 30% on average!) – so, more than half of the 'total energy flux' seems to be unaccounted for. Rather than thoroughly discussing this, the authors have decided to simply not show it in the revised version.

2) Representativeness

The energy balance related to the 'synoptic types' is assessed based on one surface energy balance station. The authors address the issue by including a short paragraph on the relative abundance of different species – and conclude that there will be 'some uncertainty' (l. 126) when applying the results of one site to the wider area of the Australian Alps. It is, however, not [only] the representativeness of the surface cover that determines the energy balance. In fact, on a 3m EB tower, the footprint (different for different wind directions – and hence synoptic conditions; but this just as an aside) of the flux measurements does hardly incorporate, the claimed percentages for different surface vegetation types.

What is relevant in complex terrain is the very local variability of the surface energy fluxes. One can measure the surface EB at a handful of sites within a few kilometers horizontal distance and one gets substantially different daily cycles for the EB components. That is, on the same day (same synoptic conditions) one site exhibits a strong daily cycle in Qh, say (resulting in a strongly positive daily sum) while a site 2 km apart with a different local slope, local exposition, 'exposure' to local flow regimes, local surface characteristics (on Fig. 2, I see many of those potentially relevant. . .), Qh starts to decrease long before local noon leading to an overall small (sometimes even negative) daily sum. Which one of the sites now produces the characteristic 'response' to the synoptic pattern? (And, more important: do those two sites show the same characteristic daily cycles on days with a given synoptic pattern? Of course, this latter question cannot be addressed with only one site – but at least it can be answered for the one site that is available – is there a characteristic daily cycle for a given synoptic pattern? In other words, is a 'median heat flux' a useful variable?).

I am not saying here that it is impossible to establish the surface EB terms for a region [in complex terrain] in relation to synoptic flow patterns. But I am saying that it is extremely difficult with only one station. And if only one station is available (and this can happen), the upscaling approach must be very careful and at least try to address the uncertainty involved (rather than sweeping it under the rug).

(some) Minor comments

l. 401 sentence

l. 433 first of all, Tab 2 yields 22 occurrences for T5 (not 24 as claimed), and second, this number does not seem to be very high (rank 3 out of 7, but much closer to the small end than the two really abundant). Fig. 6a seems to suggest that the large number is at least partially due to a few cases with up to 10 MJ day-1 (upper whisker).

l. 512 which has only one. . ..: T7 seems to be negative, too (Fig. 6a) in the median. . .

l. 518 . . . and synoptic patterns T3 and T4. . .: first of all, above (l. 516) the synoptic patterns associated with anti-cyclonic influence are identified as T1, T2, T4 and T7. Second, T3 and T4 do not have the largest negative energy fluxes, neither in median (Fig. 6a) nor in total amounts (Fig, 6b). Finally then, why would only T3 and T4 increase in frequency?

---

## Referee Comment (RC2) · Anonymous Referee #2 · 29 Apr 2020

General comments

This paper begins to address a significant gap in Australian snow literature by identifying the local energy balance and synoptic scale conditions under which snow melt occurs. While only one site has been analysed in this work, making boarder inferences of the region difficult, the identification of typical synoptic scale patterns associated with energy fluxes contributing to snow melt is of interest to the community. In addition, observational data such as those presented here may be useful for future modelling studies of Australian snow pack.

My major comment is with respect to the temporal resolution of the study, which will be

discussed below. I am in agreement with the other reviewer, that these results should not be overstated as they are from only one site, but considered as an important first step. Further minor comments are listed below.

Major comment:

The use of the daily scale of analysis, while a practical measure, may be hiding some interesting diurnal cycle features. The authors have shown that short and long wave radiation play an important role in the energy fluxes calculated, both of which by nature, have strong diurnal cycles. Some sub-daily analysis, exploring the diurnal cycles of energy fluxes for the different synoptic types may be of interest to see how different fluxes, which over the sum of the day may (or may not) balance each other out, play different roles in snowpack characteristics. In addition, sub-daily knowledge of energy fluxes would be important for evaluation of any high-resolution modelling study attempting to study snowpack in the future.

In this vein, the authors have defined a day as the period '00Z-23.59Z'. Would not have converting the UTCZ time into a 24hour period more closely aligned with the local diurnal cycle have been better? For example, when considering local meteorological effects associated with the diurnal cycle, such as anabatic or katabatic winds, which may have an important influence on local energy fluxes?

Minor comments:

Introduction in general: Some of the snowfall and weather/climate literature presented in the introduction is somewhat out of date. For example, on line 64, the Hennessy et al (2008) study has been cited, when more recent work is available in Di Luca et al. (2018).

Similarly, the studies relating to SAM and the sub-tropical ridge are quite old, with much more literature available relating climate drivers and synoptic types to southeast Australian precipitation, including discussions on how these are changing. Of note,

the authors spend some time discussing the SAM, but then go on to state that SAM accounts for relatively small variability. So perhaps the climate/weather discussion needs to be rephrased to be more specific to the local area (see Pepler et al. 2015, or Fiddes et al. 2015). In addition, I think that you should make clearer how weather types that you have identified here, eg the passage of fronts, or high pressure systems, are changing/expected to change (see Pepler et al. 2019 and Catto et al. 2014). This will give the last sentence of your abstract a bit more context and to make the importance of this study clearer in your discussion.

Line 115: I think the BoM 2018b reference is missing

Lines 119-123: I think this section about the types of vegetation would fit better under Line 112.

Lines 182:188: I'm unsure if selecting just one timestep is a good representation of cloud cover for the day. I know you mention wishing to avoid short-lived clouds, but surely even short-lived clouds have some impact on the energy balance? The himawari data should allow you to get a daily average of cloud fraction. Alternatively, providing a sub-daily analysis would resolve this too.

Line 308:314: In the discussion of T6, you state that the passage of a trough has developed into a weak lee side cyclone. Have you considered or checked that it could also be a cyclone with east coast low characteristics? I.e not associated with westerly flow? Fiddes et al. 2015 found these types of synoptic systems had some influence on extreme precip in the region.

Lines 359-365: I think this paragraph needs a bit of context. I was quite confused as to its relevance to the paper before I got nearer to the end.

Line 475-476: Re: the ground energy fluxes. Would it be possible to look at these with a seasonal perspective, to see if they play a greater role early or late in the season? This could tie in nicely with the previous findings of shorter duration of snowpack.
Figure 5 and Figure 6: Please describe figures in full in the caption. Also, it would be beneficial to use the same colour scheme for each synoptic type throughout and also avoid the rainbow colour scheme at all costs (for our colour blind colleagues!).

Catto, J. L., Nicholls, N., Jakob, C., & Shelton, K. L. (2014). Atmospheric fronts in current and future climates. Geophysical Research Letters, 41(21), 7642–7650. https://doi.org/10.1002/2014GL06194

Di Luca, A., Jason, L., & Fei, P. E. (2018). Australian snowpack in the NARCliM ensemble : evaluation , bias correction and future projections. Climate Dynamics, 51(1), 639–666. https://doi.org/10.1007/s00382-017-3946-9

Fiddes, S. L., Pezza, A. B., & Barras, V. (2015). Synoptic climatology of extreme precipitation in alpine Australia. International Journal of Climatology, 35(2), 172–188. https://doi.org/10.1002/joc.3970

Pepler, A. S., Trewin, B., & Ganter, C. (2015). The influences of climate drivers on the Australian snow season The influences of climate drivers on the Australian snow season. Australian Meteorological and Oceanographic Journal, 65(JANUARY), 195–205.

Pepler, A., Hope, P., & Dowdy, A. (2019). Long-term changes in southern Australian anticyclones and their impacts. Climate Dynamics, 53(7–8), 4701–4714. https://doi.org/10.1007/s00382-019-04819-9
* * *

---

## Author Comment (AC1) · 25 May 2020

This comment is in response to the comments made by the two reviewers on our manuscript "Quantifying the impact of synoptic weather types and patterns on energy fluxes of a marginal snowpack". We would like to thank the anonymous reviewers for their time and effort developing suggestions on how to improve our manuscript.

This comment will address the suggestions made by the reviewers, changes that we intend to make to the manuscript, and/or justification for our initial approaches. Our responses to the reviewers' comments, including our proposed changes to the manuscript, are in blue text.

Regards,

Andrew Schwartz and on behalf of co-authors

**Reviewer 1 Comments-**

General considerations

This is a resubmission of a paper with a similar title and focus, for which I have provided a quite detailed review on the previous version. The authors have made quite some changes (improvements!) with respect to the first version (data post processing and gap filling), but in my view have not been able to appropriately address the truly major concerns: the representativeness of one surface energy balance (EB) site and the treatment of EB (non-)closure. The authors have decided to 'avoid' the problem (e.g., by no more showing/discussing the evidence with respect to EB closure) or to 'downplay' it (the representativeness issue). I have two major comments making my points below. My third major concern on the first version had been the identification of synoptic patterns on a daily basis: here, the authors have added two sentences defending their choice – which is fine in principle. However, I still think that the authors miss out some potential or, in other words, introduce some unnecessary variability by choosing a not optimal reference time scale.

We thank reviewer 1 for their time, efforts and quality feedback in reviewing this paper across multiple iterations. Their suggestions have indeed improved the paper. We thank them for the positive response to most of the changes made on previous version of the manuscript.

We address the remaining concerns in our detailed response below, but it is worth stating that the previous efforts to address these concerns were not efforts to avoid or downplay these issues, but rather were our understanding or interpretation of the point being made by the reviewer (in their original review) and the corresponding corrections. It has been useful to have this additional context in the new review to better understand the issue and perspective of the reviewer, particularly in relation to the three main uses: 1) Energy Balance closure; 2) site representativeness and 3) application of daily-scale synoptic data.

1. We address the first in providing a deeper analysis and detail of energy balance closure, while accepting that there will always be uncertainty with energy balance closure, particularly over snow (see R1MC1).
2. The additional detail provided in the most recent review has led us to a slightly different appreciation of the point that reviewer 1 was making regarding site representativeness. Our original interpretation was one that the reviewer was asking for us to make a better case to justify that the study site had a footprint representative

of the study region over which we use synoptic data (this is how we addressed this criticism originally). Given the new information in this review, it appears that the point is more related to philosophical question of scale – that is: how do you on one hand use synoptic data covering by definition a large area, and then on the other make interpretation in relation to energy balances that are, by necessity, collected as single site flux measurements. We agree and accept this point, but also agree on the value and important contribution of single-site studies. As such, we add further detail and take care to edit and moderate language throughout the revised manuscript to reflect this point on scale and moving between single-site energy measurements and trying to understand synoptic drivers, see R1MC2.

3. The third main point is in relation to daily time-steps in synoptic data and the selected reference time frame (the starting point of each day, and Reviewer 2 makes some related comments that we also address below). This is a constraint for any study that must apply a seemingly arbitrary time-step definition, and we provide a justification to this decision (see R2MC1). We selected a daily period as this is the most commonly-used time-step to capture the periodicity over which synoptic-scale events impact a region. Selecting a reference time frame (start of the synoptic day) of 00 UTC (10am local) approximates the timing of the local daily rainfall benchmark (9am). This also allows our related work to take findings of this paper and apply them to a more detailed understanding of regional hydroclimate and synoptic drivers of rainfall and runoff.

Major comments
**R1MC1:** Energy balance closure
The authors have added an additional version of the surface energy balance equation, which, at least formally, addresses the non-closure and introduces the residual ($Q_{res}$).

→On l. 221, the authors claim that '$Q_{res}$ calculation and comparisons of snow pack energy flux terms were performed using the terms in eq. (2)'. This equation contains a 'energy balance closure term' ($Q_{ec}$). This term, however, is not available from the measurements. How did the authors make those 'calculation an comparisons'? (note that the non closure is not just the sum of the 5 measured terms – because it also includes the $Q_{res}$ (i.e., the energy available for melt and internal [in the snow pack] energy storage).

→Furthermore, when presenting the results, the 'ec-term' is not shown (and therefore not discussed) – of course, this is no wonder when it was not measured and cannot be derived from the measured terms. What is presented in the results section is the 'total net energy flux' (Section 3.2.4) – but it is not mentioned how this was determined: sum of the 5 measured ($Q^*$, $Q_e$, $Q_h$, $Q_g$, $Q_r$) as in eq (1) [and called $Q_m$]? At least, when comparing Fig. 6a and Table 2 (the entry for $Q_m$), one gets the impression that it is indeed $Q_m$ what is now called 'the total energy flux'.

→finally, in Table 2, $Q_m$ is listed, even if on l. 210 it is stated that $Q_m$ can be more accurately expressed as $Q_{res}$. . .

So, overall it appears that the authors have, basically, added a new equation (which is never used thereafter), do not discuss the issue, and still present the same data – and now seem to call it 'the total energy flux' instead of 'melt energy'. It is, unfortunately, so that the residual is also 'energy flux' – simply not accounted for in the form of the terms in eq (1) [it is local advection, flux divergence, storage, . . .]. This is not a subtlety. In the first version the authors had a short discussion on the energy balance closure (some 30% on average!) – so, more than half of the 'total energy flux' seems to be unaccounted for. Rather than thoroughly discussing this, the authors have decided to simply not show it in the revised version.

We agree that energy balance closure needs to be addressed in a comprehensive manner within the revised manuscript. While measurements of internal snowpack processes weren't available during the study, an updated energy balance closure approximation has been calculated using changes in snow depth and average density data at the site. By comparing the energy required for a measured reduction in snowpack snow water equivalent (SWE) during melt periods to the measured energy fluxes, energy balance closure and Qec were approximated. Mean energy balance closure during the seasons was found to be 62%, which is similar to closure over a snowpack that was found by Welch et al. (2016). The mean closure would suggest a Qec term of 38% that is not accounted for within the measurements. Though this method offers a good approximation of closure, it is affected by wind-driven snow scouring as the energy balance "truth" is calculated as a result of snow melt/removal. Therefore, mean energy balance closure has been determined for each of the synoptic types and compared to average wind characteristics.

The listing of Qm in Table 2 rather than Qres was an oversight when updating the manuscript.

**Manuscript changes:**
1. Add a paragraph to the results and discussion sections discussing energy balance closure during the study and resulting Qec. Information on average values for each synoptic type will be included and compared to the mean wind characteristics of each type to illustrate uncertainty in calculation of closure and Qec for each type.
2. Update Table 2 with correct Qres term and replace other mentions of Qm and 'total net energy flux' with Qres to avoid confusion.

**R1MC2:** Representativeness
The energy balance related to the 'synoptic types' is assessed based on one surface energy balance station. The authors address the issue by including a short paragraph on the relative abundance of different species – and conclude that there will be 'some uncertainty' (l. 126) when applying the results of one site to the wider area of the Australian Alps. It is, however, not [only] the representativeness of the surface cover that determines the energy balance. In fact, on a 3m EB tower, the footprint (different for different wind directions – and hence synoptic conditions; but this just as an aside) of the flux measurements does hardly incorporate, the claimed percentages for different surface vegetation types.

What is relevant in complex terrain is the very local variability of the surface energy fluxes. One can measure the surface EB at a handful of sites within a few kilometres horizontal distance and one gets substantially different daily cycles for the EB components. That is, on the same day (same synoptic conditions) one site exhibits a strong daily cycle in Qh, say

(resulting in a strongly positive daily sum) while a site 2 km apart with a different local slope, local exposition, 'exposure' to local flow regimes, local surface characteristics (on Fig. 2, I see many of those potentially relevant. . .), Qh starts to decrease long before local noon leading to an overall small (sometimes even negative) daily sum. Which one of the sites now produces the characteristic 'response' to the synoptic pattern? (And, more important: do those two sites show the same characteristic daily cycles on days with a given synoptic pattern? Of course, this latter question cannot be addressed with only one site – but at least it can be answered for the one site that is available – is there a characteristic daily cycle for a given synoptic pattern? In other words, is a 'median heat flux' a useful variable?).

I am not saying here that it is impossible to establish the surface EB terms for a region [in complex terrain] in relation to synoptic flow patterns. But I am saying that it is extremely difficult with only one station. And if only one station is available (and this can happen), the upscaling approach must be very careful and at least try to address the uncertainty involved (rather than sweeping it under the rug).

The authors agree that upscaling the energy balance of a single site study to the wider region should be done carefully and that spatial uncertainty needs to be addressed when any in-situ measurements are analysed. However, we also agree with reviewer 2's comment that "… these results should not be overstated as they are from only one site, but considered as an important first step." The aim of this paper is to give a first indication of the effects of standard synoptic patterns on the marginal snowpack energy balance of the Snowy Mountains. It is not intended to provide a comprehensive overview of energy balance in the region as it pertains to synoptic patterns.

**Manuscript changes:**
1. Add a paragraph in the discussion to make it clear that the energy balance fluxes measured at the site are representative of the Pipers Creek Catchment headwaters and are not intended to be 'upscaled' to be representative of the entire region. In the same paragraph, add discussion that complex terrain will contribute to variability in the measured fluxes at different locations.
2. Add additional discussion to section 2.1 "Study site and climate" that addresses spatial variability of fluxes in complex terrain and the limitations of a single-site study.

(some) Minor comments

**R1C1:** l. 401 sentence

**Manuscript change:** Fix typo in sentence on Line 401.

**R1C2:** l. 433 first of all, Tab 2 yields 22 occurrences for T5 (not 24 as claimed), and second, this number does not seem to be very high (rank 3 out of 7, but much closer to the small end than the two really abundant). Fig. 6a seems to suggest that the large number is at least partially due to a few cases with up to 10 MJ day-1 (upper whisker).

We thank the reviewer for identifying the typo in the text at Line 433 that should be 22 occurrences rather than 24. We agree that the number of T5 occurrences is relatively lower

than those of T3 and/or T7 and that the wording should be changed to better reflect the number of occurrences. We also agree with the reviewer in their assessment that the larger number is partially responsible to a few days with large energy fluxes. However, median T5 energy flux and IQR is substantially higher than any of the other types, which led to the initial wording of the sentence.

**Manuscript change:** Change number of occurrences to 22 in Line 433 and include additional detail on the distribution of T5 Qres.

**R1C3:** l. 512 which has only one. . ..: T7 seems to be negative, too (Fig. 6a) in the median. . .

You are correct, the beginning of the pattern (T7) is also negative.

**Manuscript change:** Change Line 512 to reflect T7 also being negative in its median energy flux to the snowpack.

**R1C4:** l. 518 . . . and synoptic patterns T3 and T4. . .: first of all, above (l. 516) the synoptic patterns associated with anti-cyclonic influence are identified as T1, T2, T4 and T7. Second, T3 and T4 do not have the largest negative energy fluxes, neither in median (Fig. 6a) nor in total amounts (Fig, 6b). Finally then, why would only T3 and T4 increase in frequency?

We agree that this sentence is worded poorly and needs to be changed to better convey its meaning.

**Manuscript change:** Remove the second half of the Line 516 "…and synoptic patterns T3 and T4, which have the largest negative snowpack energy fluxes, would increase in frequency" as it was initially written in a confusing manner.

**Reviewer 2 Comments-**

General comments
This paper begins to address a significant gap in Australian snow literature by identifying the local energy balance and synoptic scale conditions under which snow melt occurs. While only one site has been analysed in this work, making boarder inferences of the region difficult, the identification of typical synoptic scale patterns associated with energy fluxes contributing to snow melt is of interest to the community. In addition, observational data such as those presented here may be useful for future modelling studies of Australian snow pack. My major comment is with respect to the temporal resolution of the study, which will be discussed below. I am in agreement with the other reviewer, that these results should not be overstated as they are from only one site, but considered as an important first step. Further minor comments are listed below.

Major comment:
**R2MC1:** The use of the daily scale of analysis, while a practical measure, may be hiding some interesting diurnal cycle features. The authors have shown that short and long wave radiation play an important role in the energy fluxes calculated, both of which by nature, have strong diurnal cycles. Some sub-daily analysis, exploring the diurnal cycles of energy fluxes for the

different synoptic types may be of interest to see how different fluxes, which over the sum of the day may (or may not) balance each other out, play different roles in snowpack characteristics. In addition, sub-daily knowledge of energy fluxes would be important for evaluation of any high-resolution modelling study attempting to study snowpack in the future.

In this vein, the authors have defined a day as the period '00Z-23.59Z'. Would not have converting the UTCZ time into a 24hour period more closely aligned with the local diurnal cycle have been better? For example, when considering local meteorological effects associated with the diurnal cycle, such as anabatic or katabatic winds, which may have an important influence on local energy fluxes?

While higher-temporal resolution would result in greater detail in the diurnal patterns of the energy fluxes, daily analysis was chosen as synoptic weather, by definition, occurs on scales greater than or equal to one day. As such, the use of "days" for analysis of synoptic weather is common in research on precipitation in the Snowy Mountains region (Theobald et al., 2016;Theobald et al., 2015;Chubb et al., 2011;Fiddes et al., 2015) and glacier and snowpack energy balance (Neale and Fitzharris, 1997;Hay and Fitzharris, 1988). The chosen time step allows for the identification of common synoptic weather patterns across winter seasons and their influences on snowpack. However, we understand that examination of snowpack fluxes at smaller time-scales is also an important pursuit and refer to the work of Bilish et al. (2018) who identified diurnal patterns in fluxes at a collocated site. Further work on spatial and temporal variability of snowpack fluxes in the region is currently under review at a different journal as well.

We made a very deliberate choice with the reference time-frame, to start our "synoptic day" at 00 UTC, or 10am local. Firstly, this is the time that corresponds to daily rainfall measurements that are made at 9am local and, therefore, 00 UTC (10am local) represents the time of the synoptic data that allows us to align the synoptic profiling to daily rainfall. While this is not an explicit focus of this paper, it was in earlier "sister" papers (Theobald et al., 2016;Theobald et al., 2015) and indeed is a significant focus of the broader project funding of this work, which is to link these processes together in relation to a more detailed understanding of regional hydroclimate. Secondly, the local reference time-frame doesn't affect the analysis of the energy balance fluxes as the synoptic conditions represented in the reanalysis data would have the same effect on terrain-induced flows regardless of whether they occurred in the same local day.

Manuscript change: Add additional justification for the use of UTC 'days' as periods of analysis at Line 175.

Minor comments:
R2C1: Introduction in general: Some of the snowfall and weather/climate literature presented in the introduction is somewhat out of date. For example, on line 64, the Hennessy et al (2008) study has been cited, when more recent work is available in Di Luca et al. (2018).

Similarly, the studies relating to SAM and the sub-tropical ridge are quite old, with much more literature available relating climate drivers and synoptic types to southeast Australian

precipitation, including discussions on how these are changing. Of note, the authors spend some time discussing the SAM, but then go on to state that SAM accounts for relatively small variability. So perhaps the climate/weather discussion needs to be rephrased to be more specific to the local area (see Pepler et al. 2015, or Fiddes et al. 2015). In addition, I think that you should make clearer how weather types that you have identified here, eg the passage of fronts, or high pressure systems, are changing/expected to change (see Pepler et al. 2019 and Catto et al. 2014). This will give the last sentence of your abstract a bit more context and to make the importance of this study clearer in your discussion.

We thank the reviewer for their suggestion on updating and improving the introduction to better represent the focus of the manuscript and the impacts associated with changes to frequency of the identified synoptic types. We agree with the changes that they suggested and will incorporate them into the manuscript.

**Manuscript change:** Update introduction to include more recent literature and better illustrate expected changes to synoptic types identified in the manuscript.

**R2C2:** Line 115: I think the BoM 2018b reference is missing

**Manuscript change:** Add BoM 2018b reference.

**R2C3:** Lines 119-123: I think this section about the types of vegetation would fit better under Line 112.

Agreed, both sections are about vegetation and should be together.

**Manuscript change:** Move lines 119-123 to under Line 112.

**R2C4:** Lines 182:188: I'm unsure if selecting just one timestep is a good representation of cloud cover for the day. I know you mention wishing to avoid short-lived clouds, but surely even short-lived clouds have some impact on the energy balance? The himawari data should allow you to get a daily average of cloud fraction. Alternatively, providing a sub-daily analysis would resolve this too.

While we agree that short-lived clouds still have an impact on radiative transfer, the cloud analysis time (03Z) was chosen to develop an understanding of the broad-scale effects of each synoptic type on cloud cover and radiative transfer in the region with minimal local effects. However, it is possible that the individual timestep misrepresented the average cloud cover for the day. As such, we suggest confirming cloud cover characteristics for each day by examining the higher resolution satellite imagery.

**Manuscript change:** Confirm cloud cover classifications for days and update manuscript where necessary.

**R2C5:** Line 308:314: In the discussion of T6, you state that the passage of a trough has developed into a weak lee side cyclone. Have you considered or checked that it could also be a cyclone with east coast low characteristics? I.e not associated with westerly flow? Fiddes et

al. 2015 found these types of synoptic systems had some influence on extreme precip in the region.

We agree that it's important to distinguish between normal lee-side cyclone development and East Coast Lows as the latter produce intense winds and precipitation that the reviewer noted. All days that were classified as T6 were either lows that had moved into the area through normal progression of synoptic patterns or were the result of troughs developing low characteristics as they moved into the region.

**R2C6:** Lines 359-365: I think this paragraph needs a bit of context. I was quite confused as to its relevance to the paper before I got nearer to the end.

We agree that this paragraph seems a bit out of place and irrelevant in its current position and without more context.

**Manuscript change:** Add context to the results about the relevance of calculating transition probabilities.

**R2C7:** Line 475-476: Re: the ground energy fluxes. Would it be possible to look at these with a seasonal perspective, to see if they play a greater role early or late in the season? This could tie in nicely with the previous findings of shorter duration of snowpack.

We thank the reviewer for your suggestion to conduct analysis on the seasonality of Qg fluxes. Analysis had been conducted to determine if Qg was higher at the beginning of the seasons due to high soil temperatures that had not yet reached a winter "equilibrium". No patterns existed in Qg during the 2016 winter season and slightly higher Qg values that did exist at the beginning of 2017 were found to be within one standard deviation of the mean Qg values from 2016 and no clear seasonal trend was noted.

**R2C8:** Figure 5 and Figure 6: Please describe figures in full in the caption. Also, it would be beneficial to use the same colour scheme for each synoptic type throughout and also avoid the rainbow colour scheme at all costs (for our colour blind colleagues!).

Thank you for your suggestion to avoid rainbow colour schemes as it wasn't something that we had considered, we will make sure that it is changed to eliminate problems for those that might be colour blind.

**Manuscript change:** Change Figure 5 colour scheme to match Figure 6 and elaborate on captions for Figures 5 & 6.

**References provided by Reviewer 2:**
Catto, J. L., Nicholls, N., Jakob, C., & Shelton, K. L. (2014). Atmospheric fronts in current and future climates. Geophysical Research Letters, 41(21), 7642–7650. https://doi.org/10.1002/2014GL06194

Di Luca, A., Jason, L., & Fei, P. E. (2018). Australian snowpack in the NARCliM ensembleâ˘A´r: evaluation, bias correction and future projections. Climate Dynamics, 51(1), 639–666. https://doi.org/10.1007/s00382-017-3946-9

Fiddes, S. L., Pezza, A. B., & Barras, V. (2015). Synoptic climatology of extreme precipitation in alpine Australia. International Journal of Climatology, 35(2), 172–188. https://doi.org/10.1002/joc.3970

Pepler, A. S., Trewin, B., & Ganter, C. (2015). The influences of climate drivers on the Australian snow season. Australian Meteorological and Oceanographic Journal, 65(JANUARY), 195– 205.

Pepler, A., Hope, P., & Dowdy, A. (2019). Long-term changes in southern Australian anticyclones and their impacts. Climate Dynamics, 53(7–8), 4701–4714. https://doi.org/10.1007/s00382-019-04819-9

**Author Response References:**
Bilish, S. P., McGowan, H. A., and Callow, J. N.: Energy balance and snowmelt drivers of a marginal subalpine snowpack, Hydrol Process, 32, 3837-3851, 2018.

Chubb, T. H., Siems, S. T., and Manton, M. J.: On the Decline of Wintertime Precipitation in the Snowy Mountains of Southeastern Australia, J Hydrometeorol, 12, 1483-1497, 10.1175/Jhm-D-10-05021.1, 2011.

Fiddes, S. L., Pezza, A. B., and Barras, V.: Synoptic climatology of extreme precipitation in alpine Australia, International Journal of Climatology, 35, 172-188, 2015.

Hay, J. E., and Fitzharris, B. B.: The synoptic climatology of ablation on a New Zealand glacier, Journal of Climatology, 8, 201-215, 10.1002/joc.3370080207, 1988.

Neale, S. M., and Fitzharris, B. B.: Energy balance and synoptic climatology of a melting snowpack in the Southern Alps, New Zealand, International Journal of Climatology, 17, 1595-1609, 10.1002/(SICI)1097-0088(19971130)17:14<1595::AID-JOC213>3.0.CO;2-7, 1997.

Theobald, A., McGowan, H., Speirs, J., and Callow, N.: A Synoptic Classification of Inflow-Generating Precipitation in the Snowy Mountains, Australia, J Appl Meteorol Clim, 54, 1713-1732, 10.1175/Jamc-D-14-0278.1, 2015.

Theobald, A., McGowan, H., and Speirs, J.: Trends in synoptic circulation and precipitation in the Snowy Mountains region, Australia, in the period 1958-2012, Atmos Res, 169, 434-448, 10.1016/j.atmosres.2015.05.007, 2016.

Welch, C. M., Stoy, P. C., Rains, F. A., Johnson, A. V., and McGlynn, B. L.: The impacts of mountain pine beetle disturbance on the energy balance of snow during the melt period, Hydrol Process, 30, 588-602, 10.1002/hyp.10638, 2016.

---

## Author Response (AR2)

**Point-by-point response to reviewer comments including author responses and changes made to the manuscript**

This comment is in response to the additional comments made by reviewer #2 on our manuscript "Quantifying the impact of synoptic weather types and patterns on energy fluxes of a marginal snowpack". We would like to extend our sincere gratitude to the reviewer for their additional comments and suggestions on how to improve the manuscript.

This comment will address the suggestions made by the reviewer and changes that we made to the manuscript in response. Our responses to the reviewers' comments, including our proposed changes to the manuscript, are in blue text.

Regards,

Andrew Schwartz and on behalf of co-authors

**Reviewer 2 Comments-**

The authors have significantly updated the literature as suggested, and further justified their analysis choices. I think this paper provides an important reference for future Australian snowpack studies but I note that the broader applicability of the paper must not be overestimated. I have a couple of minor corrections listed below.

Abstract + Conclusion: I suggest that you mention/highlight that only one measurement site is used in the abstract and conclusion and caveat the broader applicability of this work. I realise this is done in the the Discussion, but we all know most people only read the abstract and conclusions, and hence this fact should be made readily available.

We agree that the information needed to be made more available in the paper and have added the suggested changes.

**Manuscript change:** Added sentences to the abstract (Lines 19-20) and conclusion (Lines 600-602) specifying that the study used a single-site and caution should be used before upscaling.

Conclusion, Line 623: Please reference papers showing the trends of frontal systems for this region.

**Manuscript change:** Added citations of papers showing trends of frontal systems in southeast Australia.

Figure 3 still contains a rainbow colour scheme which should be changed.

**Manuscript change:** Changed colour of Figure 3c to remove rainbow colour scheme.

[revised manuscript text omitted]